# Stratifying ALS Patients by Mode of Inheritance Reveals Transcriptomic Signatures Specific to sALS and fALS

**DOI:** 10.3390/ijms26189234

**Published:** 2025-09-22

**Authors:** Alexandria Awai, Erica L. Johnson, Tiandong Leng, John Patrickson, Michael C. Zody, James W. Lillard

**Affiliations:** 1Department of Microbiology, Biochemistry & Immunology, Morehouse School of Medicine, Atlanta, GA 30310, USA; aawai@msm.edu (A.A.);; 2Department of Neurobiology, Biochemistry & Immunology, Morehouse School of Medicine, Atlanta, GA 30310, USA; 3New York Genome Center, New York, NY 10013, USA

**Keywords:** ALS, sALS, fALS, WGCNA, DESeq2, GO, neurodegeneration, GWAS

## Abstract

Amyotrophic lateral sclerosis (ALS) is a terminal neurodegenerative disease, marked by considerable clinical and molecular heterogeneity. While several genetic drivers have been linked to familial ALS (fALS), the biology of sporadic ALS (sALS)—which accounts for the majority of ALS cases—remains poorly defined. To address this gap, we analyzed 247 bulk mRNA-sequenced post-mortem tissue samples from the lumbar spinal cord and motor cortex and compared expression profiles between fALS, sALS, and controls. Variance-stabilized DEGs from DESeq2 analysis were used as inputs for weighted gene co-expression network analysis (WGCNA). Finally, gene ontology was used to identify transcriptomic signatures and biological pathways unique to sALS and fALS. In the spinal cord, sALS samples exhibited specific downregulation of mitochondrial complex I subunits (e.g., NDUFS8 and NDUFB7) and regulatory genes (e.g., AURKAIP1 and ATP5F1D), suggesting compromised metabolic resilience. In the motor cortex, a co-expression module associated with adaptive immune function and leukocyte infiltration was downregulated in sALS yet upregulated in fALS, indicating distinct inflammatory pathways between these two forms of ALS. Together, our findings highlight that while sALS and fALS are largely the same disease, they exhibit distinct transcriptomic signatures. By accounting for mode of inheritance in study designs—particularly sALS, which represents ~90% of ALS cases—researchers may reveal deeper insights into ALS pathology. This perspective could enable more targeted therapeutic strategies, ultimately improving outcomes for all ALS patients.

## 1. Introduction

Amyotrophic Lateral Sclerosis (ALS), also known as Lou Gehrig’s disease, is a progressive neurodegenerative disorder affecting upper and lower motor neurons that control voluntary movement via the cortico-spinal and bulbar tracts [1]. This devastating condition leads to progressive muscle weakness, atrophy, and finally paralysis, including respiratory insufficiency and death. The National ALS Registry reported approximately 33,000 cases of ALS in the US in 2022, and by 2030, these cases are estimated to rise by more than 10% to over 36,000 cases [2,3,4]. This disease, with its intricate nature and diverse manifestations, poses significant challenges to the medical and research communities. It also imposes substantial annual costs in the US ranging from $212 million to $1.4 billion USD, covering direct and indirect healthcare resource costs [4,5,6].

ALS cases can be broadly categorized into two types: familial ALS (fALS), which accounts for approximately 5–10% of cases, and sporadic ALS (sALS), comprising the remaining 90–95%. Assignment to either group is based on the hereditary nature of the causative genetic mutation. Currently, causal genetic mutations are primarily associated with fALS [7]. Many pathway alterations associated with ALS-linked mutations occur in all patients (pALS); however, the exact genetic causes of sporadic ALS remain unclear—a key focus of ongoing research [8].

Currently, four genes—C9ORF72, SOD1, FUS, and TARDPB—are the most common monogenic causes of familial ALS (fALS), though many other genes have been linked to the disease [9]. Hexanucleotide repeat expansions in a non-coding region of C9orf72 are predominantly seen in ~40% of fALS cases and 5–10% sALS, as well as ALS with frontotemporal dementia (FTD) [10]. In addition to monogenic inheritance, oligogenic and polygenic models of ALS have emerged [11]. In oligogenic models, the presence of two or more risk variants can lead to disease onset [12]. In contrast, in polygenic models, numerous single-nucleotide polymorphisms (SNPs) are present and increase risk of interactions with environmental hazards [13].

Protein misfolding and aggregation in motor neurons are trademarks of ALS etiology [14]. Variants in the gene encoding TAR-DNA-binding protein (TDP-43) result in the presence of cytosolic misfolded aggregates [15]. FUS, which resembles TDP-43 in function and pathology, accumulates abnormally in the cytoplasm, forming granules and aggregates after nuclear exclusion [16]. Finally, mutations in copper–zinc superoxide dismutase 1 (SOD1) can lead to decreased or increased function, resulting in reduced enzyme activity and the accumulation of toxic free radicals that damage DNA and protein misfolding [17]. Additionally, mutant SOD1 can accumulate inside the mitochondrial intermembrane space (IMS) or deposit on the outer mitochondrial membrane (OMM), which leads to mitochondrial dysfunction and motor neuron pathology [18].

Immune dysfunction has been established as part of the etiology of ALS; whether immune dysfunction is a cause of ALS or a response to stressors is still being investigated. However, mutations linked to ALS can lead to inefficient immune responses and/or the accumulation of cellular damage, suggesting that an underlying innate immunodeficiency may also contribute to the pathogenesis of ALS [19]. The activation of microglia and astrocytes characterizes neuroinflammation. This immune dysregulation may allow for the entry of leukocytes and antibodies into the central nervous system, further contributing to ALS pathology [20].

ALS research and care have made significant strides, yet challenges persist that impact early diagnosis and equitable access to treatment. The absence of universally accepted biomarkers complicates early and accurate diagnosis, often leading to delays that affect treatment efficacy [21]. The disease’s heterogeneous presentation further complicates timely identification, especially in atypical cases [22]. These factors underscore the importance of ongoing research into gene–environment–time interactions and the development of large, diverse cohorts to enhance our understanding of ALS etiology and to inform more inclusive and effective therapeutic strategies [23,24].

As a rare disease, each ALS patient offers valuable insight into the complex etiology of ALS. Researchers have acknowledged the necessity of a broader, more cohesive approach to ALS—one that fosters collaboration, diversification, and standardized methodologies among clinicians and investigators (https://www.targetals.org accessed on 1 January 2025; https://www.iamals.org accessed on 1 January 2025). An example of these efforts is Target ALS, an initiative aimed at expanding clinical and preclinical testing sites, providing resources for high-impact research, and developing cohesive collection protocols. The ALS Toolkit, an electronic health record-based system, aims to standardize data capture during clinical encounters, supporting both patient care and research [25]. The ALS Natural History Consortium is creating a comprehensive dataset to reflect the full ALS population, revealing greater heterogeneity compared to clinical trial data [26]. Given ALS’s molecular heterogeneity—particularly the poorly defined biology of sALS, which accounts for ~90% of cases—we performed transcriptomic analyses of post-mortem motor cortex and lumbar spinal cord, stratified by mode of inheritance, to uncover distinct and shared molecular pathways that can inform subtype-specific biomarkers and therapies.

## 2. Results

### 2.1. DESeq2

Both sALS and fALS were compared to the same tissue-specific non-neurological controls. The gg4way plot overlays the log-fold change values from the sALS versus control and fALS versus control contrasts with an adjusted *p*-value < 0.05 (Figure 1A and Figure 2A). This unified approach allows for comparison of gene expression between the two ALS subtypes relative to the control group, highlighting both common and unique expression patterns.

In the MC, expression changes were observed in sALS and fALS. In fALS, 2823 genes were upregulated, and 4003 genes were downregulated (padj < 0.05). Stringent shrinkage dispersion parameters (s < 0.005) highlighted 1 upregulated and 8 downregulated genes. In sALS, 2387 genes were upregulated and 3067 genes were downregulated, with shrinkage parameters (s < 0.005); 7 genes were upregulated, and 3 were downregulated. Of those, 2 genes had similar expression changes between the cohorts MTND1P23 and MTCO1P12. A literature search of DEGs revealed 6 genes previously reported to be associated with ALS, while 11 DEGs may suggest novel associations with ALS (Figure 1B).

In the LSC, expression changes were observed in both sALS and fALS. 5285 genes were upregulated, and 5116 genes were downregulated in fALS cases (padj < 0.05), when adjusted for shrinkage dispersion (s < 0.005), 51 were upregulated and 6 were downregulated. In sALS, 3465 upregulated and 3911 downregulated genes were observed. When adjusted for shrinkage dispersion, 11 were upregulated and 2 were downregulated. Of the shrinkage dispersion results, 7 genes had similar expression changes amongst the cohort, 5 were upregulated (LYZ, CHIT1, GPNMB, DNASE2B, and CCL18) and 2 were downregulated (CXCL11, MTND1P23). A literature search of LSP DEGs revealed 20 genes that were previously associated with ALS, while the remaining 44 have not been previously reported in association with ALS and are denoted in bold font in Figure 2B.

### 2.2. Weighted Gene Coexpression Analysis and Gene Ontology

WGCNA networks were built from the same variance-stabilized expression matrix used in our DEG analysis; consequently, module “hub” genes need not rank among the top individual DEGs. Instead, each module reflects a coordinated pattern of expression changes that signal pathway-level disruption.

Motor Cortex. Of the 21 WGCNA motor-cortex modules (mcMods), 3 modules were significant (*p* < 0.05) and associated with sALS, fALS, or both (*p* < 0.05) (Table 1). mcMods 9 and 2 were adjacent in the WGCNA dendrogram (Figure 3), denoting their relatedness to one another; collectively, these modules highlight upregulation of nuclear and intracellular structural processes. mcMod 5, by contrast, was characterized by coordinated down-regulation of genes involved in synaptic signaling. Taken together, these shared mcMods indicate a common ALS signature: upregulation of nuclear-structure pathways alongside suppression of synaptic function.

Additionally, the binary matrix allowed for further exploration of phenotypes associated with the modules (Appendix A). mcMod 9 tracked a mid-life onset profile and intermediate disease duration: it was upregulated in individuals with symptom onset in the 50–59 age range and with 2–5 years of disease duration. mcMod 9 also showed a sex effect (upregulated in females, downregulated in males) and was upregulated across both fALS and sALS strata, consistent with a signal present in ALS irrespective of inheritance.

mcMod 2 displayed a related, but more clinically acute pattern: upregulation of eigengenes in shorter duration bins (< 2 years and 2–5 years), upregulation in females, both fALS and sALS cases. Together, M2 and M9 point to cortex modules that scale with disease presence and course.

In contrast, mcMod 5 showed the opposite sex trend (upregulated in males, downregulated in females), and eigengenes were downregulated in cases of mid-life onset (50–59) and 2–5 years duration. Eigengenes were also downregulated in cases of limb onset, and in both sALS and fALS cases.

While mcMod18 was linked to both sALS and fALS, they displayed opposing expression patterns: upregulated in fALS and downregulated in sALS (Figure 3A). In addition, eigengenes were upregulated in females and downregulated in males (Appendix A). This module comprises genes involved in immune and metabolic functions, notably MHC-II antigen presentation and complement activation (Table 2).

Lumbar Spinal Cord. Seven of twenty-one lumbar spinal-cord modules (scMods) were significant (*p* < 0.05) and associated with sALS, fALS, or both (Table 3). Six scMods were significantly associated with both sALS and fALS and had similar expression patterns. scMods 1 and 17, adjacent in the dendrogram (Figure 3B), highlighted disrupted cytoplasmic translation and protein aggregation. Downregulated scMod 14 (tau-binding proteins), scMod 12 (synaptic-plasticity regulation), and scMods 3 and 10 (axon–glial support) together suggest dampened synaptic and axon-maintenance pathways. Lastly, one module was sALS-specific: downregulated scMod 13, enriched for mitochondrial NADP-dependent metabolic processes.

When looking at the complete heatmap, additional phenotypic associations can be divulged (Appendix A). scMod 1 was upregulated and associated with 2–5 years of disease duration across limb and bulbar onset and in both fALS and sALS, indicating a general ALS-tracking signal.

scMod 17 was the most broadly upregulated module, associated with limb (and axial + limb) onset, fALS, sALS, 2–5 years duration (also evident in <2 years), and mid-life onset (50–59), with an additional upregulation in younger age at death (50–59) and a tail in 80–89 onset. This robust presentation across multiple phenotypes is expected since this module relates to protein aggregation, a key manifestation of ALS.

In contrast, scMod 3 showed consistent downregulation and association with limb onset, sALS, fALS, 2–5 years duration, and mid-to-late onset ages (50–59, 60–69). scMod 10 likewise was downregulated and associated with limb onset, sALS, fALS, 2–5 years duration, and mid-life onset bins (with a further association at >5 years duration), indicating another pathway that is suppressed in prototypical ALS features.

scMod 12 and scMod 14 showed similar profiles: Both were downregulated in cases of limb onset, sALS, fALS, 2–5 years duration, and mid-life onset (50–59), but were upregulated among individuals with age at death 70–79, suggesting that eigengenes are dampened in active disease yet enriched in those who reach older age at death.

Finally, scMod 13 was downregulated in cases with onset in the 40–49 age range, very late age at death (90–99), and sALS, indicating a subtler pathway tied to an age-defined strata. Exact effect sizes and FDR values for each association are provided in the supplemental tables; phenotypes containing “N.A.” or “Not applicable” were excluded from all summaries above, as these are controls (Appendix A).

## 3. Discussion

Our study highlights the complex morphology of ALS. The heterogeneity of ALS and limited biomarkers compound the challenges in diagnosis. Our initial hypothesis focused on identifying distinct expression changes between fALS and sALS cases. By stratifying patients in this way, we identified genes that have not been reported in the context of ALS in both sALS and fALS cases, as well as DEGs that are specific to each condition (Figure 1A and Figure 2A). In the MC analysis, we identified an adaptive immune network that, while associated with both sALS and fALS, may function in opposing ways in each phenotype. In the LSC analysis, we identified metabolic gene networks uniquely related to the sALS phenotype. Overall, the gene networks associated with either fALS or sALS revealed deficits in neuronal structure, transport, metabolic balance, immune infiltration, and decreased synaptic integrity that converge on neuronal vulnerability in ALS.

Subcategories for ALS based on differing pathology in motor neurons of the cortex have been discussed [27], including oxidative and proteotoxic stress, activate glial cells, high levels of retrotransposon expression, and TARDBP/TDP-43 dysfunction.

### 3.1. Gene Ontology and Interpretation of Modules Specific to sALS

WGCNA of the MC identified Module 18, which contains transcripts significantly downregulated in sALS but upregulated in fALS compared to controls. Although the module itself did not reach strict significance (*p* = 0.09), it showed distinct associations with sALS (*p* = 0.05) and fALS (*p* = 0.02), suggesting divergent regulatory roles in these subtypes. Key eigengenes within this module (Table 2) include immune- and microglia-related genes such as ITGB2, TYROBP, C3, ITGAM, SPI1, and SYK, alongside metabolic regulators (e.g., SLC7A7) and cytoskeletal interactors (NCKAP1L, LCP1). Gene ontology analysis of motor cortex (MC) tissue revealed that Module 18-associated genes were enriched for adaptive immune responses, neutrophil degranulation, intercellular communication, and MHC class II antigen presentation. Hub genes further underscored this adaptive immune signature, including integrin subunit beta 2 (ITGB2), NCK-associated protein 1-like (NCKAP1L), complement component C3 (C3), the immune signaling adaptor TYROBP, and lysosomal protein LAPTM5.

Protein–protein interaction networks were generated via the STRING database (Figure 4A) and revealed that many of these hub genes are functionally interconnected, with dark gray edges indicating high-confidence interactions (scores ≥ 0.7) and lighter edges representing medium-confidence (scores ≥ 0.4). Hub genes in this network are predominantly associated with immune and metabolic processes relevant to ALS pathology. STRING associations highlight previously studied protein–protein interactions and warrant further functional investigation.

WGCNA of LSC samples identified scMod 13 (*p* = 0.02) downregulated and significantly correlated with sALS phenotype relative to controls. Eigengenes within this module include: ARL6IP4, ADRM1, SCAND1, AURKAIP1, PGLS, NDUFS8, ATP5F1D, ALKBH7, MZT2B, DDX54, RBM42, MRPS34, BAD, COMT, COPE, MPG, GADD45GIP1, CCDC124, CORO1B, NDUFB7, MICOS13, MVB12A, and NME3 (Table 2). Protein–protein interaction networks generated via the STRING database reveal functionally interconnected hub genes (Figure 4B). Several hub genes displayed high-confidence interactions (scores ≥ 0.7), including NDUFS8 and NDUFB7, subunits of mitochondrial Complex I, as well as ATP5F1D, a component of mitochondrial ATP synthase, and MRPS34, a mitochondrial ribosomal protein. These associations underscore the involvement of mitochondrial function and energy metabolism in sALS pathology. Additional hub genes, such as AURKAIP1 (linked to proteolysis), ALKBH7 (involved in programmed necrosis and metabolic processes), and GADD45GIP1 (associated with cell cycle regulation), suggest dysregulation of pathways implicated in protein turnover and cellular stress responses.

### 3.2. Differential Expression Analysis Revealed Novel Genes Enriched in sALS and fALS

The expression of distinct mitochondrial pseudogenes, MTCO1P12, MTCO2P12, MTND1P23, was enriched in ALS samples; MTCO1P12 was downregulated in both fALS and sALS (fALS: padj = 2.3 × 10^−12^, s = 6.83 × 10^−7^; sALS: padj = 2.66 × 10^−7^, s = 0.0001). Although labeled ‘pseudogenes’, transcripts such as MTCO1P12, MTCO2P12, and MTND1P23 can act as regulatory ncRNAs that modulate the stability or translation of their protein-coding paralogues; their downregulation therefore signals potential impairment of cytochrome-c-oxidase activity and oxidative phosphorylation.

Two additional transcripts linked to stress and innate immunity were upregulated in sALS cases, lung cancer-associated transcript 1 (LUCAT1; padj = 3.56 × 10^−6^, s = 0.003) and 2′-5′-Oligoadenylate Synthetase-Like (OASL; padj = 0.001, s = 1.55 × 10^−31^), reinforcing a theme of inflammatory signaling within cortical tissue. In fALS cases, Selectin E (SELE; padj = 0.0002, s = 2.3 × 10^−8^) was upregulated, also suggesting an inflammatory response in ALS.

In the LSC, lipid-handling and extracellular-matrix genes topped the novel list. Apolipoprotein C1 (APOC1; padj = 1.08 × 10^−21^, s = 3.65 × 10^−10^), known for binding phosphatidylcholine and blocking phospholipase activity, and previously implicated in Alzheimer’s disease, was selectively upregulated in fALS, hinting at a conserved lipid-detox response across neurodegenerative contexts. Deoxyribonuclease 2β (DNASE2B; fALS: padj = 1.98 × 10^−12^, s = 1.85 × 10^−20^; sALS: padj = 1.25 × 10^−5^, s = 1.52 × 10^−3^), an acid-active endonuclease, was upregulated in both disease groups, possibly reflecting increased clearance of damaged DNA within the inflammatory milieu. Matrix remodeling was underscored by the elevation of Vitrin (VIT; padj = 6.03 × 10^−5^, s = 1.89 × 10^−3^) in sporadic cases, while an immune component surfaced through immunoglobulin λ constant 2 (IGLC2; padj = 9.32 × 10^−1^, s = 2.02 × 10^−4^), which was likewise upregulated in sporadic ALS.

Collectively, these novel DEGs expand the lipid-metabolism, mitochondrial, and immune-matrix narratives already emerging from our dataset and provide a focused shortlist for mechanistic follow-up in both cortical and spinal compartments. Together, the highlighted DEGs point to four tightly linked processes uniting familial and sporadic ALS: mitochondrial bioenergetic stress, phospholipid detox and membrane repair, innate-immune activation, and extracellular-matrix remodeling, framing metabolic strain and neuroinflammation.

## 4. Methods and Materials

### 4.1. Data Acquisition and ALS Consortium

Amyotrophic lateral sclerosis (ALS) and control samples were provided by the New York Genome Center (NYGC) Center for Genomics of Neurodegenerative Disease and TargetALS. NYGC’s collaborative approach enlisted the aid of multiple institutions to form the ALS Consortium. The Consortium’s collaborative network of 45 institutions worldwide includes Sample Collection Sites and Specimen Collection Sites distributed across various cities and countries. This partnership of clinicians, basic scientists, geneticists, and computational biologists establishes a framework for applying whole genome sequencing and functional omics to ALS research (ALS Consortium). These sites serve as crucial points for collecting biological samples and specimens, enabling a comprehensive study of ALS across diverse populations. All post-mortem tissue samples were obtained with informed consent from the donors or their legal representatives, and study protocols were approved by the Institutional Review Boards of the participating institutions within the New York Genome Center’s ALS Consortium. Generation of genomics libraries from postmortem material in the NYGC ALS Consortium is covered by IRB# BRANY-15–08-292–385. All materials have been anonymized, were obtained postmortem, and are therefore considered “not human subjects” [27,28].

### 4.2. Data Management, Processing, Quality Control, and Verification

All New York Genome Center/ALS Consortium specimens pass a QC pipeline spanning sample receipt through data release. Total RNA was isolated from flash-frozen tissue (TRIzol, Invitrogen, Thermo Fisher Scientific, Waltham, MA, USA; chloroform, Sigma-Aldrich, St. Louis, MO, USA; QIAGEN RNeasy, QIAGEN GmbH, Hilden, Germany.), quality-checked on an Agilent Bioanalyzer, and 500 ng was used to build rRNA-depleted, strand-specific libraries (KAPA Stranded RNA-seq + RiboErase; NEXTflex indexes, PerkinElmer, Austin, TX, USA). Pooled libraries (≈375 bp inserts) were sequenced on Illumina HiSeq 2500 or NextSeq (2 × 125 bp) (Illumina, San Diego, CA, USA), generating ≈40–50 million paired reads per sample [27,28].

Reads from samples with RIN ≥ 5.5 were aligned to the hg38 human genome using STAR v2.7.6.55, allowing for a 4% mismatch rate and up to 100 alignments per read to ensure capture of young transposon sequences. The abundance of gene and transposon sequences was calculated using TEtranscripts v2.2.3 [28,29].

### 4.3. Cohort Filtering

ALS cases and non-neurological controls underwent analysis across two distinct brain regions—the motor cortex and lumbar region of the spinal cord (Table 1). Filtering criteria were consistently applied. The entirety of the ALS cohort included symptomatic and non-symptomatic patient samples, ALS-FTD, and neurological controls; only samples of confirmed ALS cases, FTD excluded, with family history identifiers and non-neurological controls, were used. The familial-versus-sporadic designation was based exclusively on clinical pedigree assessment recorded at enrollment and supplied in the consortium metadata, ensuring robust stratification of the cohort.

It is important to note that while sALS makes up about 90% of ALS cases, it represents fewer than 50% of the samples in the cohort. The reasons for this vary, from delayed to missed diagnosis for sporadic cases to a lack of SALS enrollment in clinical trials where data collection measures are taken. Tissue specificity was addressed by analyzing motor cortex (MC) and lumbar spinal cord (LSC) samples separately. All analyses were conducted independently for each tissue type. Additional information on mutations present and sex is available in Appendix A.

### 4.4. Differential Gene Expression Analysis

The experimental design investigated differential gene expression (DEG) between control, sALS, and fALS LSC and MC samples. Low-count genes (< 10) were filtered out of total feature counts (*n* = 19,548), resulting in 18,597 in LSC and 18,278 in MC features for analysis. DESeq2, an R package Version 2024.12.1+563 for analyzing high-throughput RNA-seq count data, was used to identify significantly up- or downregulated genes compared to controls. A DESeq2 data object was constructed from the read count data and metadata using “FamHistory” as the primary design factor, enabling assessment of multiple contrasts, including “sALS vs. Control” and “fALS vs. Control”, where controls are non-neurological.

Significant DEGs were defined by an adjusted *p*-value (padj) below 0.05, applying the Benjamini–Hochberg correction, and an absolute log2 fold change greater than 1. This approach ensured rigorous detection of differential expression while maintaining a balanced rate of discovery. Additionally, effect size estimation with apeglm [29] provides stringent empirical Bayes shrinkage parameters to DESeq2 results to reduce noise and allow true differences to be represented. Summary statistics of the counts of upregulated and downregulated genes were reported. Variance stabilizing transformation (vst), a parametric dispersion fit, was used to approximate variance-stabilized log2-scale counts while correcting for size factors to be used for downstream clustering analysis. DEGs were visualized in gg4way plots; sALS and fALS results under apeglm parameters were highlighted via volcano plots. Finally, strip plots were employed to confirm the effectiveness of the experimental design by illustrating known DEGs among ALS subtypes and controls.

#### Differential Expression Design

Equation (1): The comparison factor in design, FamHistory, includes three groups—sALS, fALS, and Control-NA, with Control-NA serving as the reference leveldds.Cortex = DESeqDataSetFromMatrix (countData = ctsMatrix,colData = metaTissue,design = ~FamHist)dds.Tissue <– DESeq (object = dds.Tissue).(1)

### 4.5. Binary Construction

Metadata was converted to categorical formats for variable grouping. Age of onset was divided into several ranges to facilitate analysis. These ranges were defined as follows: under 40 years, 40–49 years, 50–59 years, 60–69 years, and 70 years and above. Similarly, the age at death was categorized into ranges to provide a structured view of patient outcomes. These categories were as follows: under 50 years, 50–59 years, 60–69 years, 70–79 years, and 80 years and above. To better understand the progression of the disease, the duration was grouped into three categories: Less than 2 years, 2–5 years, and more than 5 years. This categorization allows for analyzing short-term, medium-term, and long-term disease trajectories.

Lastly, the site of motor onset was categorized based on the primary affected area. The categories included bulbar (affecting speech and swallowing), upper limb, lower limb, and other/mixed. This classification aids in understanding the diseases’ initial manifestation and their potential impact on patient outcomes. These groupings allow for more effective analysis of trends and patterns across different subsets of patients, facilitating the identification of possible correlations between clinical features and gene expression patterns in the subsequent WGCNA. Binary indicators were created for phenotypes—Sex, Family History, Age of Onset, Age at Death, Disease Duration, and Site of Motor Onset. This allowed for effective integration with gene expression data and the identification of relationships between clinical features and gene expression patterns using WGCNA.

### 4.6. Correlation Analysis Using WGCNA

The weighted gene co-expression network analysis (WGCNA) R software (v4.2.2) package was used to assess gene co-expression profiles across ALS samples. Variance-stabilized expression profiles derived from the DESeq2 analysis served as the input for WGCNA, linking single-gene differential signals to higher-order co-expression modules whose hub genes represent key drivers of ALS pathology. Module eigengenes were the key outputs from WGCNA [30,31]. Eigengenes, or hub genes, representing each module were assessed for correlation to sample-specific disease-related traits of interest [31]. This formed a framework for downstream analysis of gene networks of disease relevance.

This workflow began with quality control, where genes and samples with excessive missing values were flagged. The data were then processed blockwise, with pairwise gene correlations calculated using robust methods (such as bicor, with a fallback to Pearson when necessary). Biweight midcorrelation (bicor) is used to provide vigorous correlation with less weight given to outliers [29,30]. Bicor and utilizing transformed data are crucial for summarizing correlation in transcriptomic data, which can display a dynamic range, resulting in high variance across samples. These correlations were transformed into an adjacency matrix via soft thresholding to approximate a scale-free topology. The adjacency matrix was further refined into a Topological Overlap Matrix (TOM), which considered both direct interactions and shared neighbors between genes, setting the stage for hierarchical clustering.

Following clustering, modules of co-expressed genes were identified using dynamic tree-cutting methods. Eigengenes, the first principal components representing each module’s dominant expression pattern, were computed for all modules. Gene expression was then assessed for module membership (kME), and those with low kME were either removed or reassigned to more appropriate modules. Finally, modules with highly similar eigengenes, indicating similar expression profiles, were merged based on a predefined similarity threshold. This systematic process ensured that the resulting modules represent robust, biologically meaningful groups of co-expressed genes that could be correlated with external traits for further analysis. Results were visualized using various functions from WGCNA, including plotDendroAndColors, plotEigengeneNetworks, and labeledHeatmap. Parameters used were as follows: power = 8, deepSplit = 4, minModuleSize = 100, mergeCutHeight = 0.15, TOMdenom = “mean”, corType = “bicor”, networkType = “signed”, pamStage = TRUE, pamRespectsDendro = TRUE, maxBlockSize larger than the number of genes being clustered (30,000), and reassignThresh = 0.05.

### 4.7. Gene Ontology (GO)

Gene Ontology (GO) was performed using Eric Dammer and Divya Nandakumar’s GO program in RStudio version 12.1+563. Modules with significant MEs based on Kruskal–Wallis test were analyzed and discussed (*p* < 0.05), as well as modules that displayed expression changes between sALS and fALS. The top five biological pathways were ranked based on false discovery rate (FDR) and fold enrichment score > 20. Under this guideline, one cortical module and one spinal cord module were analyzed using GO.

## 5. Conclusions

In a chronic inflammatory and complex disease setting like ALS, it might seem contradictory to observe downregulated genes associated with immune response. However, multiple mechanisms can converge to produce this result (Figure 5). For example, C3 downregulation in sALS spinal cord may reflect adaptive or protective feedback to counter sustained activation. In the early disease stage, C3 might have been upregulated, but over time (or in certain CNS regions), the system shifts to a lower-expression state, potentially as a compensatory effort to protect neurons from ongoing complement-mediated damage.

In addition to the immune response observed, our findings further support metabolic dysfunction as a result of proteotoxic or oxidative stress in ALS pathology (Figure 5). WGCNA of LSC samples revealed two genes encoding subunits of mitochondrial protein ubiquinone oxidoreductase (Complex I)—NDUFS8 and NDUFB7—downregulated in sALS samples. Other genes implicated in mitochondrial regulation and cell cycle function (AURKAIP1, ATP5F1, MRPS34, GADD45GIP1, ADRM1) also showed reduced expression and were associated with sALS, suggesting a broader pathway disruption. Whether this dysfunction is causal or secondary to oxidative stress and its specific cell type prevalence remains to be determined.

Our work reflects that sporadic and familial ALS share a core signature of glial activation and neuronal dysfunction but diverge in specific mitochondrial and immune-modulatory pathways. By stratifying patients by mode of inheritance for transcriptomic analysis, we uncovered novel candidate genes, such as MTCO1(P12), VIT, and APOC1, along with pathway-level disruptions that suggest that while being the same disease, there are gene signatures that vary between sALS and fALS.

## Figures and Tables

**Figure 1 ijms-26-09234-f001:**
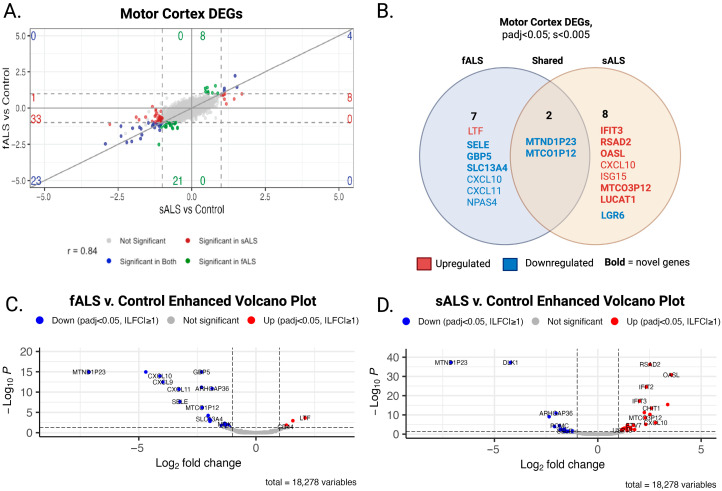
Integrated transcriptomic comparison of sporadic and familial ALS in the motor cortex. (**A**) Four-way scatter (gg4way) plot of variance stabilized transformed data comparing log_2_ fold-changes (LFC) for each gene in sALS vs. control (x-axis) and fALS vs. control (y-axis). Blue = significant in both contrasts; red = significant in sALS; green = significant in fALS; gray = not significant. Dashed lines mark |LFC| = 1 and adjusted *p* = 0.05 boundaries. Genes in the upper-right and lower-left quadrants represent shared pathological signals, whereas off-diagonal genes highlight subtype-specific changes. (**B**) Venn diagram of differentially expressed genes (DEGs) unique to fALS (**left**), unique to sALS (**right**), and shared (**center**). Red labels = upregulated; blue labels = downregulated. Genes in bold have not been previously reported in association with ALS, underscoring their novelty. (**C**) Volcano plot—fALS vs. control (**D**) Volcano plot—sALS vs. control. Points represent individual genes; red = highlight DEGs with |LFC| ≥ 1; FDR < 0.05; s < 0.005. Top-ranked novel genes from panel b are annotated for visibility.

**Figure 2 ijms-26-09234-f002:**
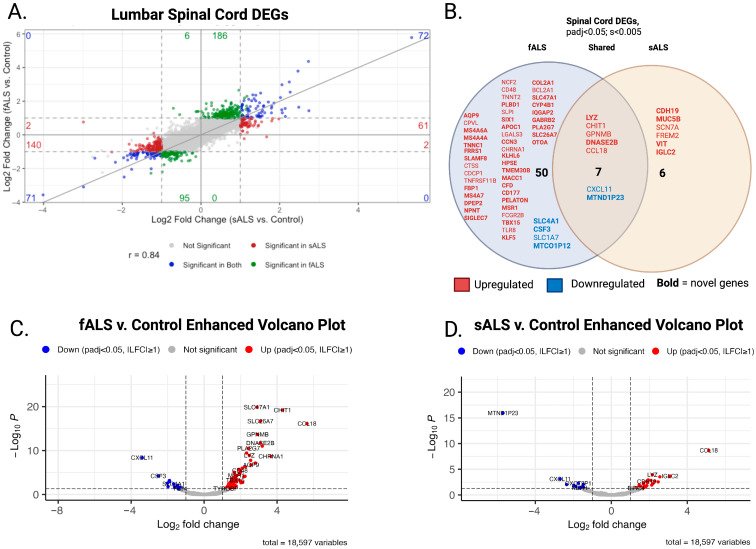
Integrated transcriptomic comparison of sporadic and familial ALS in spinal cord samples (**A**) Four-way scatter (gg4way) plot of variance stabilized transformed data comparing log_2_ fold-changes (LFC) for each gene in sALS vs. control (x-axis) and fALS vs. control (y-axis). Blue = significant in both contrasts; red = significant in sALS; green = significant in fALS; gray = not significant. Dashed lines mark |LFC| = 1 and adjusted *p* = 0.05 boundaries. Genes in the upper-right and lower-left quadrants represent shared pathological signals, whereas off-diagonal genes highlight subtype-specific changes. (**B**) Venn diagram of differentially expressed genes (DEGs) unique to fALS (**left**), unique to sALS (**right**), and shared (**center**). Red labels = upregulated; blue labels = downregulated. Genes in bold have not been previously reported in association with ALS, underscoring their novelty. (**C**) Volcano plot—fALS vs. control and (**D**) Volcano plot—sALS vs. control. Points represent individual genes; red = highlight DEGs with |LFC| ≥ 1; FDR < 0.05; s < 0.005. Top-ranked novel genes from panel b are annotated for visibility. Together, the four panels illustrate (i) the degree of concordance between sporadic and familial ALS transcriptional changes, (ii) subtype-specific gene signatures, and (iii) a set of previously unreported candidate genes that warrant follow-up functional studies.

**Figure 3 ijms-26-09234-f003:**
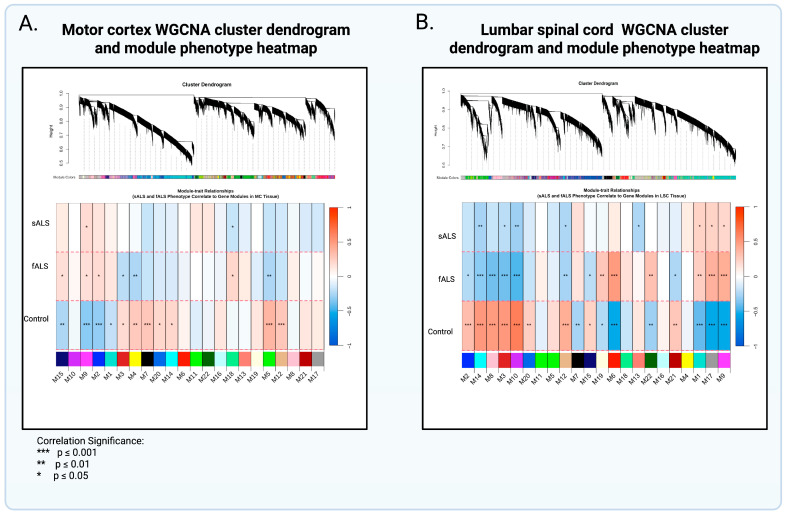
WGCNA Cluster Dendrogram and Module-Phenotype Heatmap. The cluster dendrogram depicts the strength of the eigengene association to each module. In the first row below the dendrogram, each colored vertical streak represents a gene with membership in the module of that color, which contains a group of highly coexpressed transcripts. The heatmaps display the relationship between gene networks (modules) and phenotypes (sALS/fALS/control). Overlaid asterisks denote the Student’s *p*-values for bicor significance of trait correlation to the module eigengenes (*p* ≤ 0.05 = *, *p* ≤ 0.01 = **, *p* ≤ 0.001 = ***). Module–Trait bicor color scale (−1, blue; 0, white; +1, red). (**A**) mcModule 18 was upregulated in fALS and downregulated in sALS. (**B**) spM13 was downregulated in sALS alone, suggesting the occurrence of sporadic-specific transcriptional changes.

**Figure 4 ijms-26-09234-f004:**
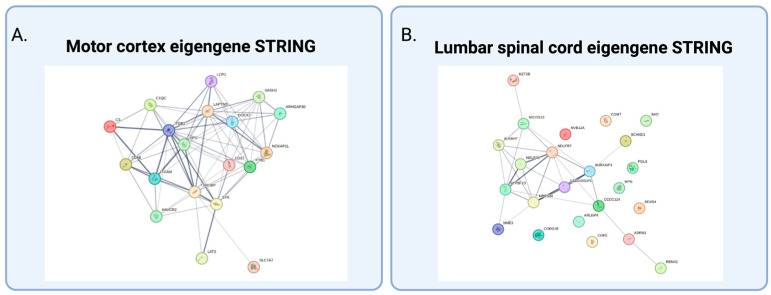
Eigengene STRING Network Analysis. Nodes depict encoded protein, while edges represent protein interactions. Dark gray edges indicate high-confidence interactions (scores ≥ 0.7) and lighter edges represent medium-confidence (scores ≥ 0.4). (**A**) mcMod 18; (**B**) scMod 13.

**Figure 5 ijms-26-09234-f005:**
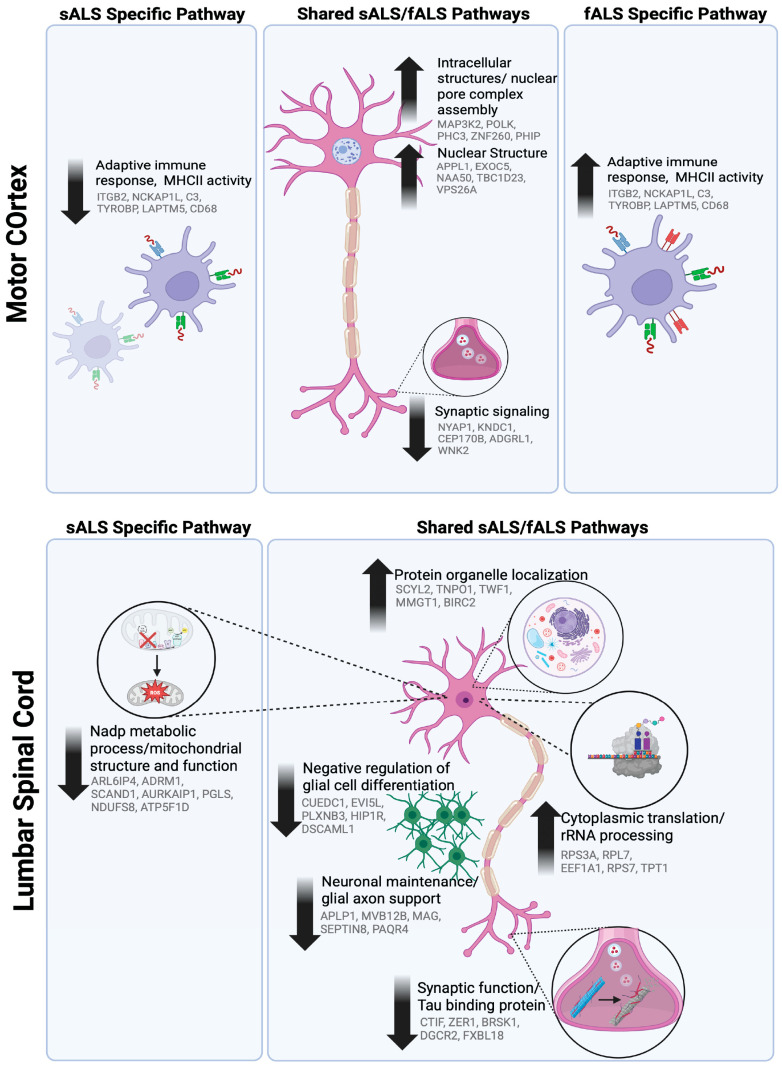
Tissue- and genotype-specific pathway dysregulation in ALS. RNA-seq counts were variance-stabilized in DESeq2 and utilized for weighted gene co-expression network analysis (WGCNA). Module eigengenes significantly associated with sporadic ALS (sALS), familial ALS (fALS), or both were functionally annotated with STRING and summarized here. Top panels (Motor Cortex); bottom panels (Lumbar Spinal Cord). Icons depict the principal cellular compartment or process affected; eigengenes within each pathway are displayed. Created with BioRender.com.

**Table 1 ijms-26-09234-t001:** WGCNA motor cortex modules.

	Motor Cortex Module	Gene Ontology	Hub Genes	MEs ANOVA *p*-Value:
Shared Gene Networks				
Upregulated				
	mcM9	Intracellular structures/nuclear pore complex assembly	MAP3K2; POLK; PHC3; ZNF260; PHIP	0.02
	mcM2	Nuclear structure	APPL1; EXOC5; NAA50; TBC1D23; VPS26A	0.05
Downregulated	mcM5	Synaptic signaling	NYAP1; KNDC1; CEP170B; ADGRL1; WNK2	0.05
Upregulated in fALS Downregulated in sALS	mcM18	Immune and metabolic processes; MHCII and compliment system	ITGB2, NCKAP1L, C3, TYROBP, LAPTM5, CD68	0.09

**Table 2 ijms-26-09234-t002:** WGCNA lumbar spinal cord modules.

	Spinal Cord Module	Gene Ontology	Hub Genes	MEs ANOVA *p*-Value:
Shared Gene Networks				
Upregulated				
	scM1	Protein organelle localization	SCYL2; TNPO1; TWF1; MMGT1; BIRC2	0.02
	scM17	Cytoplasmic translation/rRNA processing	RPS3A; RPL7; EEF1A1; RPS7; TPT1	0.04
Downregulated	scM14	Synaptic function/Tau binding protein	CTIF; ZER1; BRSK1; DGCR2; FXBL18	0.01
	scM3	Negative regulation of glial cell differentiation	CUEDC1; EVI5L; PLXNB3; HIP1R; DSCAML1	0.05
	scM10	Neuronal maintenance/ glial-axon support	APLP1; MVB12B; MAG; SEPTIN8; PAQR4	0.01
	scM12	Regulation of neuronal synaptic plasticity	NEURL1; BSN; JPH3; JPH4; IQSEC3	0.01
sALS Network				
Downregulated	scM13	Nadp metabolic process/mitochondrial structure and function	ARL6IP4; ADRM1; SCAND1; AURKAIP1; PGLS; NDUFS8; ATP5F1D	0.02

**Table 3 ijms-26-09234-t003:** Cohort configuration.

Course Region	fALS	sALS	Non-Neurological Control	Total *n*
Motor Cortex	48	31	69	148
Lumbar Spinal Cord	32	14	53	99

## Data Availability

ALS Consortium dataset: The New York Genome Center’s ALS Consortium. New York Genome Center, n.d., https://www.nygenome.org/science-technology/collaborative-research-programs/neurodegenerative-disease-research/als-consortium/ (accessed on 1 May 2024).

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
