# Peer review of "Stratifying ALS Patients by Mode of Inheritance Reveals Transcriptomic Signatures Specific to sALS and fALS"

_ijms, 2025, doi:10.3390/ijms26189234_

Round 1

Reviewer 1 Report

Comments and Suggestions for Authors

This manuscript presents a well-conducted transcriptomic analysis of postmortem tissue from ALS patients, stratified by familial (fALS) and sporadic (sALS) inheritance. Using DESeq2 and weighted gene co-expression network analysis (WGCNA), the authors identify both shared and distinct gene expression signatures in the motor cortex and lumbar spinal cord across ALS subtypes. The findings highlight key immune and mitochondrial pathways that differ between sALS and fALS, offering new insights into disease heterogeneity and potential avenues for subtype-specific biomarker or therapeutic development.

Comment 1: In Figures 1A and 2A, are the gg4way plots generated using the stringent shrinkage dispersion parameters? The plots appear to display more than ten significant genes in both fALS and sALS. Please clarify this point in the Methods or figure legend.

Comment 2: In Figure 1B (Venn diagram), MTCO1P12 is shown as downregulated in both fALS and sALS, but this conflicts with Figure 1D, where MTCO1P12 appears upregulated in the volcano plot. Please check and clarify this inconsistency.

Comment 3: In Figures 1C, 1D, 2C, and 2D (volcano plots), the upper left quadrant should correspond to downregulated genes, and the upper right quadrant to upregulated genes. The current legend is confusing: dots are marked as green for Log2FC and red for both p-value and Log2FC significance. I understand the authors wish to highlight significant genes in both measures in red; however, dots above the dashed line already indicate significant genes. I suggest recoloring the upregulated and downregulated genes in a manner consistent with Figures 1B and 2B for clarity.

Comment 4: In Figure 2B, the text states there are 46 unique genes in fALS, but the figure shows 50 unique genes—46 upregulated and 4 downregulated. Please reconcile this discrepancy.

Comment 5: For the fALS group, please provide more detailed information on the genetic backgrounds included (e.g., which known ALS-associated mutations were represented) as well as the gender distribution (male/female) and age ranges. Such details for both subtypes would improve the interpretation of the DEG analyses.

Author Response

1) gg4way + “stringent shrinkage”

Response: Thank you—we agree this needs clarification. The aplm shrinkage methods were not applied in gg4way plots (Figs. 1A, 2A). This will be better explained in the methods and legend.

Comment 2 (MTCO1P12 direction inconsistency)

Response: Please review this point again. MTCO1P12 is downregulated in both sALS and fALS. In Figure 2D (cortex DESeq results) MTCO3P12 is upregulated in sALS cases. I believe that was the confusion. A list of DEGs is provided in the supplemental data.

Comment 3 (volcano quadrant/legend clarity)

Response: Agreed. We will (i) ensure the upper-left = down-regulatedupper-right = up-regulated convention; (ii) recolor points to match the Venn diagrams (blue = downred = upgrey = not significant); and (iii) simplify the legend so that padj < 0.05 is indicated by position relative to the dashed line, with saturated color used only for significant points. We will update captions to reflect this standardized scheme across Figs. 2C–D and 3C–D (figure numbering changed in resubmission).

Comment 4 (Fig. 2B count mismatch: 46 vs 50)

Response: Thank you—we will make the text and figure agree. Unfortunately, we did not provide the correct Venn diagram in the submission; the revised version will reflect that both text and image agree. The gene list is provided in a Supplementary Table to avoid ambiguity.

Comment 5 (fALS genetic background; sex/age details)

Response: We agree these details aid interpretation. We will add a baseline characteristics table (by subtype) reporting:

  • Genetic background in fALS (counts of known ALS-associated variants present in the metadata),
  • Sex distribution (M/F)  for both fALS and sALS,

Reviewer 2 Report

Comments and Suggestions for Authors

  This is a bioinformatics-only study of raw data generated by the NYGC ALS Consortium (not clarified in the text until the Materials and Methods section).  The rationale of some of the approaches are unclear and there is a robust failure to compare the results of this study to other ALS transcriptomics studies (including those using the NYGC ALS data set).  I do not understand the rationale of using stringent shrinkage dispersion parameters to exclude thousands of identified DEGs for analysis, the vast majority of which are likely to be "real".  Similar WGCNA analyses have been done on NYGC ALS data, and it is unclear why this studies' analysis is different or more informative than these previous studies.  I am skeptical of grouping all the fALS cases together, as it seems likely that C90rf72 cases might be fundamentally different than SOD1 cases.  The Material and Methods section is both missing relevant information (did the analyses use uniquely aligned reads, excluding transposon counts that are clearly relevant to ALS?  Was strandedness accounted for in the analysis?) and includes seemingly irrelevant information (why are age ranges described in the "Binary Construction" section, when this does not appear to be relevant to the analyses presented?).  The text also contains curious and unsupported statements, e.g., "Many ALS-linked mutations occur in all patients " (pg 2 lines 57 and 58) and "MTCO1P12, MTCO2P12 and MTND1P23 can act as regulatory (pg10 line 281; there is no citation and I cannot find any mention of this in the literature).  This may be a byproduct of the submission/review process, but Figure 5 is unreadable no matter how much it is magnified.

Author Response

1) Scope not clear (bioinformatics-only; NYGC source mentioned late)

Response: We will clarify in the Abstract and Intro (first paragraph) that this is a secondary/bioinformatics analysis of NYGC ALS Consortium RNA-seq data.

2) Rationale unclear; limited comparison to prior ALS transcriptomics (incl. NYGC)

Response: We will add a Related Work/Context paragraph in the Intro and expand the Discussion to compare our findings to prior ALS transcriptomic studies (including NYGC-based analyses), reporting overlap, directionality, and pathway concordance.

3) Use of “stringent shrinkage dispersion parameters” excludes many DEGs

Response: The rationale to use more stringent filters, such as shrinkage dispersion, is to focus on larger, more stable effects. However, similar to your train of thought, the stringent parameters were not used as input for WGCNA.

4) WGCNA already done on NYGC; what’s different/novel here?

Response: We will explicitly state our novelty: (i) inheritance-stratified networks (sALS vs fALS) in both motor cortex and lumbar spinal cord samples (ii) and module–trait associations by inheritance subtype. We will add a short contrast to the prior WGCNA results in the introduction.

5) Skepticism about pooling all fALS (C9orf72 vs SOD1 likely distinct)

Response: Agreed. However, our scope of the project was to investigate any changes between familial and sporadic cases, not within familial cases. This note calls for future analysis with a larger cohort.

We will report n per mutation subgroup and caution where power is limited.

6) Methods missing details (unique reads? transposons? strandedness?)

Response: We will expand Methods to specify: aligner/tool versions and parameters; uniquely mapped primary alignments and MAPQ threshold used for counting; library strandedness; and whether TE/transposon reads were included.

7) Irrelevant “Binary Construction” info (age ranges) vs modeling covariates

Response: We have added information based on the entirety of the binary construct.

8) Unsupported statement: “Many ALS-linked mutations occur in all patients”

Response: We will delete or revise this to: “Many pathway alterations associated with ALS-linked mutations occur in all patients (pALS); however, the exact genetic causes of sporadic ALS remain unclear…” The original phrasing will be removed.

https://www.mdpi.com/2075-4426/10/3/101

9) Claim about mitochondrial pseudogenes acting as regulatory elements

Response: 

You are correct; there is currently no work that directly addresses MTCO1P12, MTCO2P12, and MTND1P23 as regulators. However, I was speaking to the idea that transcribed pseudogenes can affect functioning genes. We will soften this language to observational, signal warranted follow-up, and add references to the general point.

https://www.ncbi.nlm.nih.gov/pubmed/23829530https://www.ncbi.nlm.nih.gov/pubmed/23900003.

10) Figure 5 unreadable

Response: I did not have that issue with this figure on my end, but I will check the formatting to be sure.